# Transmission dynamics and vaccination strategies for Crimean-Congo haemorrhagic fever virus in Afghanistan: A modelling study

**Juan F. Vesga**[1,2]*, **Madeleine H. A. Clark**[3], **Edris Ayazi**[4], **Andrea Apolloni**[5,6], **Toby Leslie**[7], **W. John Edmunds**[1,2], **Raphaëlle Métras**[1,2,8]

**1** Centre for Mathematical Modelling of Infectious Diseases, London School of Hygiene & Tropical Medicine, London, United Kingdom, **2** Department of Infectious Disease Epidemiology, London School of Hygiene & Tropical Medicine, London, United Kingdom, **3** Integrated Understanding of Health, Research Strategy and Programmes, Biotechnology and Biosciences Research Council, Swindon, United Kingdom, **4** Ministry of Public Health, Massoud Square, Kabul, Afghanistan, **5** CIRAD, UMR ASTRE, Montpellier, France, **6** ASTRE, Univ Montpellier, CIRAD, INRA, Montpellier, France, **7** International Health, London, United Kingdom, **8** INSERM, Sorbonne Université, Institut Pierre Louis d'Épidémiologie et de Santé Publique (Unité Mixte de Recherche en Santé 1136), Paris, France

* juan.vesga-gaviria@lshtm.ac.uk

## Abstract

### Background

Crimean-Congo haemorrhagic fever virus (CCHFV) is a highly pathogenic virus for which a safe and effective vaccine is not yet available, despite being considered a priority emerging pathogen. Understanding transmission patterns and the use of potential effective vaccines are central elements of the future plan against this infection.

### Methods

We developed a series of models of transmission amongst livestock, and spillover infection into humans. We use real-world human and animal data from a CCHFV endemic area in Afghanistan (Herat) to calibrate our models. We assess the value of environmental drivers as proxy indicators of vector activity, and select the best model using deviance information criteria. Finally we assess the impact of vaccination by simulating campaigns targeted to humans or livestock, and to high-risk subpopulations (i.e, farmers).

### Findings

Saturation deficit is the indicator that better explains tick activity trends in Herat. Recent increments in reported CCHFV cases in this area are more likely explained by increased surveillance capacity instead of changes in the background transmission dynamics. Model-ling suggests that clinical cases only represent 31% (95% CrI 28%-33%) of total infections in this area. Vaccination campaigns targeting humans would result in a much larger impact than livestock vaccination (266 vs 31 clinical cases averted respectively) and a more effi-cient option when assessed in courses per case averted (35 vs 431 respectively). Targeted vaccination of farmers is impactful and more efficient, resulting in 19 courses per case

**Data Availability Statement:** All the data used in this study is publicly available as detailed in the text and supporting information.

**Funding:** This research is funded by the Department of Health and Social Care using UK Aid funding and is managed by the National Institute for Health and Care Research. The views expressed in this publication are those of the author(s) and not necessarily those of the Department of Health and Social Care. The funders had no role in study design, data collection and analysis, decision to publish, or preparation of the manuscript.

**Competing interests:** The authors have declared that no competing interests exist.

averted (95% CrI 7–62) compared to targeting the general population (35 courses 95% CrI 16–107)

## Conclusions

CCHFV is endemic in Herat, and transmission cycles are well predicted by environmental drivers like saturation deficit. Vaccinating humans is likely to be more efficient and impactful than animals, and importantly targeted interventions to high risk groups like farmers can offer a more efficient approach to vaccine roll-out.

## Author summary

Crimean-Congo haemorrhagic fever virus (CCHF) is an understudied emerging pathogen and the cause of increasingly frequent outbreaks of haemorrhagic fever in humans in several parts of the world. Here we bring together an important body of work in different aspects of the ecology and epidemiology of CCHF to shed light on its transmission dynamics into humans and the role of environmental drivers. These results show that over the years an endemic pattern of CCHFV transmission has been established within livestock, and the frequency of human cases mirrors the seasonal pattern of livestock transmission. Our analysis further suggests that an important fraction of cases in humans might be subclinical, and the volume of transmission into humans might be much larger than previously thought. We examine the potential impact of vaccination, which suggest that not only human vaccination could be more impactful than animal vaccination, but also that targeted strategies in human high risk groups could be very effective. Our results raise important insights for future vaccine development and important questions on the optimal conditions for conducting Phase III vaccine trials in humans.

## Introduction

Crimean-Congo haemorrhagic fever virus (CCHFV) is an emerging tick-borne zoonotic pathogen which can lead to cases of fatal haemorrhagic fever in humans. In recent years, outbreaks of CCHF in humans have increased in frequency, and the virus is now endemic in several countries in the Middle East, Africa, Asia, and Southeast Europe. The wide geographical distribution of tick species which are able to harbour the virus provides added concern that the disease may spread further afield. This, and CCHF's epidemic-proneness has led the World Health Organisation to include CCHFV in the group of priority pathogens for research and development into improved vaccines, therapeutics and diagnostics [1].

The transmission dynamics of CCHFV is complex due to the interplay between environmental factors affecting tick activity and their life cycles, the asymptomatic transmission within multiple vertebrate species (wild and livestock), and behavioural factors behind the risk of spillover into humans. *Hyalomma* spp. are the main vectors of CCHFV, and *Hyalomma marginatum* complex the most frequently associated species. Tick activity has been associated with environmental and meteorological variations, which might drive seasonal transmission patterns [2]. *Hyalomma* spp, specifically, thrives during the hot summer months in dry weathers, however its adaptability to colder temperatures has been also reported, explaining in part the expanding geographical area of influence of CCHFV [2–4]. Emergence of CCHFV has also

been linked to importation of livestock species [5] and changes in agricultural activities, which affect the habitats of intermediate hosts of CCHFV [2,6].

Despite the absence of a safe and effective licensed vaccine against CCHFV, the development of a stable animal model [7] has meant that several vaccine candidates are now being studied. Inactivated virus [8,9], DNA [10], mRNA [11], and plant-expressed glycoprotein formulations [12] amongst others, are part of the current development pipeline. Inactivated vaccines have been routinely used in humans before in Bulgaria [8], with reported reductions in incidence, but the lack of data on efficacy and safety of this formulation has prevented its wider use.

CCHFV is considered a priority emerging pathogen, but important gaps in our understanding of transmission dynamics into humans and a formal assessment of the potential impact of vaccines are still necessary to advance a global research agenda and for developing a roadmap for CCHF.

Human CCHFV cases have been reported in Afghanistan at least since 1998, first in Takhar province, and later in Herat province where most of the cases have emerged since 2002 [13]. However, in recent years the distribution of cases has extended to most provinces in the country. Neighbouring countries such as Pakistan, Iran, Turkmenistan, and Tajikistan, also report annual cases of CCHFV. These countries are located in the ecological range of activity of *Hyalomma* spp. Transboundary livestock movement is thought to aid transmission [14].

Here we present a first approach to modelling CCHFV transmission amongst livestock and from livestock into humans, in Herat, an endemic area of Afghanistan.

Herat reported CCHFV outbreaks in 2008 and 2017 with an estimated case fatality ratio (CFR) ranging from 22% to 33% in humans [15,16]. Seroprevalence studies carried out in the area have reported high IgG seroprevalence in livestock (~75%) and higher seroprevalence among humans involved in livestock activities (farming, animal husbandry, etc) [17]. High seroprevalence in livestock suggests endemic transmission in animal hosts, although further serological evidence is not available to confirm this. Furthermore, the drivers behind trends in human spillover are not fully understood and the possible seasonality driven by environmental factors has not been assessed yet.

In this work, we aim to shed light on the main factors driving CCHFV transmission in western Afghanistan and ascertain whether the disease is endemic or epidemic, as a case study for CCHFV in general. We expand this case to explore the impact of selected vaccination strategies on disease incidence and mortality reduction in humans.

## Methods

### Ethics statement

This study has obtained approval from the ethics committee at the London School of Hygiene and Tropical Medicine (Reference number: 26612). All the data used has been aggregated and anonymised.

### Mathematical model

We model CCHFV transmission in livestock, and from livestock to humans in two steps (**Fig 1**).

In step 1, we define a deterministic *susceptible-infected-recovered-susceptible* (SIRS) model structure for livestock, stratified in five yearly age groups (see **S1 Fig**). Transmission between animals occurs as a function of prevalence of infectious livestock and a driving environmental factor (e.g., saturation deficit, soil temperature) which acts as a surrogate indicator of tick

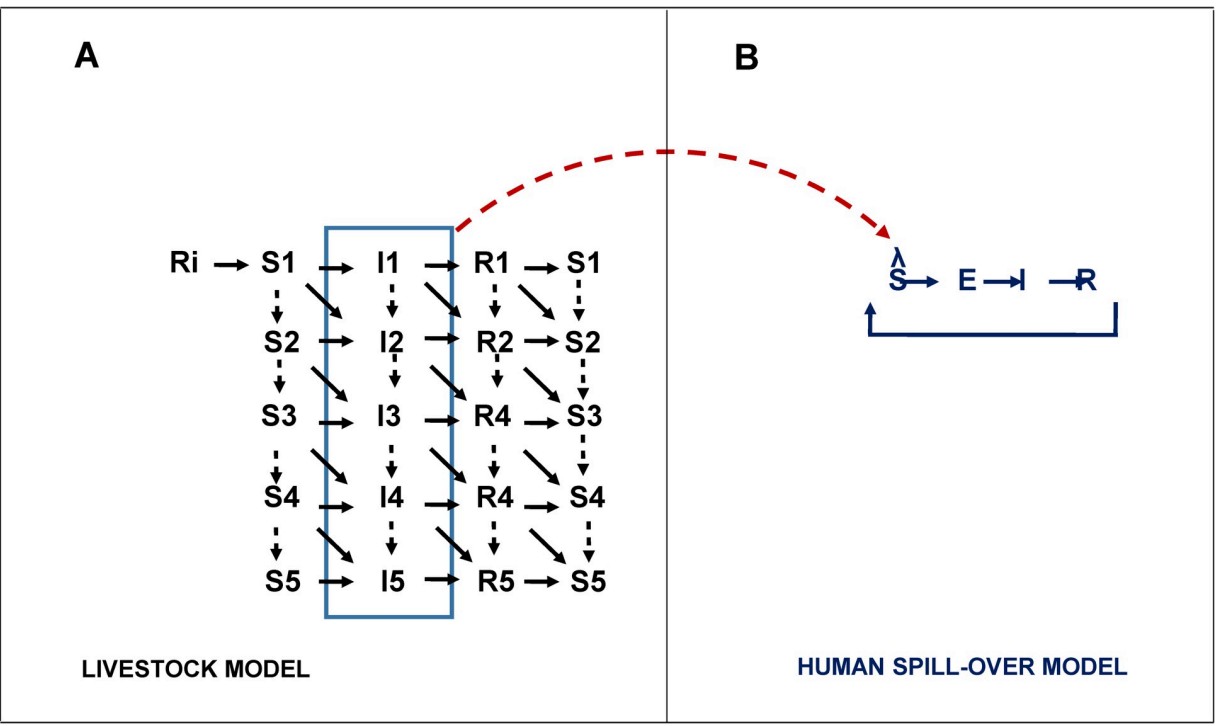

**Fig 1. Model schematic of CCHFV transmission.** We modelled CCHFV viral transmission between livestock, and from livestock to humans. In panel A, livestock were stratified in five yearly age-groups. Animals are born into the model at a rate proportional to mortality to maintain equilibrium. A fraction of livestock will acquire immunity through colostrum exposure in the first days after birth, here denoted as compartment Ri, the remaining fraction will enter the model through S1. The fraction moving to Ri is proportional to CCHFV prevalence at each time t. We assume a colostrum acquired immunity loss after six months. We model transmission between livestock with a risk function that renders the expected tick activity as a function of environmental drivers, and incorporates CCHFV infection prevalence in livestock and a scaling factor for the climatic indicator.. Infectious animals (I) recover after seven days, on average passing into the recovery compartment (R). We assume waning immunity with an average rate of 5 years$^{-1}$, hence the transition R -> S. (B), we formulate the human spillover structure as an SEIRS model, governed by a series of stochastic transition events. Transmission follows a force of infection λ, that is defined by the infectious livestock prevalence at time t $\sum_{\{a=1\}}^{5} \frac{I_a}{N_L}$ and a relative risk of transmission that conveys the differential risk by human occupation (i.e., farmers, and other). This connecting link is represented by the red dashed arrow connecting the two models. This implies a sequence of events in the runtime in which a realisation of the animal model is run over the time period, producing a vector of prevalence as output. This vector is subsequently passed as an input to the human spillover model. From the structure in panel B, is also evident that we allow a loss of acquired immunity R->S.

activity.. Hence in livestock, we define the force of infection as following,

$$\lambda_L = \beta_L \frac{\sum_a I_a}{N_L}$$

Where the term $\frac{\sum_a I_a}{N_L}$ represents the prevalence of infectious livestock at any given point in time, while $\beta_L$ is the transmission coefficient representing the combination of transmission likelihoods between tick and livestock, and tick activity. We can also write,

$$\beta_L = \frac{R_L(t)}{D_{iL}}$$

Where $R_L(t)$ is the reproduction number in livestock at each point in time $t$, and is defined as a function of the environmental driver used; $D_{iL}$ is the duration of the infectious period in livestock. Since environmental drivers are here reflecting a measure of tick activity, we incorporate the conditions that best reflect tick activity in relation to each driver. For a temperature

dependent reproduction number, for example, we describe a system where adult *Hyalomma spp*. activity occurs above 12˚C [2,18] and increases as temperature increases. Once temperature reaches above 30˚C, ticks prefer to bury into soil [19], thus we write a function for declining transmission.

In step 2, we use a stochastic *susceptible-exposed-infected-recovered-susceptible* (SEIRS) (see **S2 Fig**) model for transmission of CCHFV from livestock into humans. We define the process of transmission with a random binomial process where probability of event depends on infection prevalence in livestock at each time *t* and a risk multiplication factor to capture excess risk among farmers (assumed to be the high risk group). Human to human transmission is not modelled, assuming that this component is not relevant in sustaining CCHFV outbreaks. See **S1** and **S2 Text** for a full model description and model equations.

The model is fitted to data by calibrating relevant model parameters within a Bayesian framework (see calibration procedures in **S3 Text**). In **Table 1**, we present the list of model parameters and the calibrated values. We compare our model output against target data on age stratified CCHFV seroprevalence in livestock, risk stratified seroprevalence in humans (i.e., farmers and other occupations), and time series of reported CCHFV cases in humans (**Table A** in **S3 Text**). Calibration diagnosis can be found in **Figs A-C in S3 Text.**

**Exploring epidemiological and environmental drivers.** In the absence of tick activity data, we use the environmental factors that influence tick dynamics and its trends over the year. These factors are incorporated as drivers of transmission of CCHFV between livestock. To explore the different environmental drivers and potential epidemiological conditions that better explain the observed trends in Herat's data, we systematically compare models in two steps.

In step 1, we calibrate the models four times, each time using a different environmental driver, namely soil temperature, saturation deficit, relative humidity and normalized difference vegetation index (NDVI).

We retrieved the relevant data for the specific geographical location and time period from available sources. Each driver, with its source and relevance in tick activity can be found in **Table 2** (For further details on construction of these indicators see **S4 Text**).

In step 2, we further explore assumptions about the epidemiological factors behind the trends in reported human CCHF cases over the years. For this, we use the best model selected in step 1, and test three potential scenarios that have been hypothesised elsewhere [13,15], namely: A) Increments in CCHF reported cases reflect increment in reporting capacity (baseline assumption); B) Increased influx of livestock from other endemic regions (with a fixed reporting capacity); and, C) Increased influx of livestock from other endemic regions, and increased reporting capacity combined.

For livestock, we assume wanning immunity over an average period of 5 years after infection. This assumption allows us a better model calibration when compared against the lifelong immunity assumption (see **S3** and **S4 Figs**).

In both steps 1 and 2, the most appropriate model is selected using a Deviance Information Criterion (DIC) approach.

**Vaccination strategies.** We expand the model structure presented above to incorporate and test the impact of different vaccination strategies in the model (see S1 and S2 Figs for full model structure description). Using the best calibrated model as baseline we introduce four vaccination scenarios, where we combine different levels of vaccine coverage among livestock and humans as well as frequency of campaign roll-out. As follows,

a. 80% of livestock in a single campaign approach

b. 80% of livestock yearly

**Table 1. Model parameters.**

| Parameter description | Notation | Input Values/Estimated* | Source |
|---|---|---|---|
| Natural history of disease | | | |
| *Livestock* | | | |
| Duration of infectiousness in livestock | $D_{iL}$ | 7 days | Gonzalez et al., 1998[20] |
| Duration of colostrum acquired immunity (months) | $D_{aL}$ | 8.3 (CrI 95% 2–10) | Estimated |
| Mean time to loss of immunity in adult livestock (months) | $D_{mL}$ | 52 (CrI 95% 46–76) | Estimated |
| Proportion of livestock immune at time 0 by age¥ group $a$ | $R_a(t)$ | $R_a(t) = \begin{cases} 0.29 & for\, a=1 \\ 0.48 & for\, a=2 \\ 0.8 & for\, a=3 \\ 0.87 & for\, a=4 \\ 0.87 & for\, a=5 \end{cases}$ | Barthel et al., 2014[21] |
| *Humans* | | | |
| Duration of latent period in humans | $D_{lH}$ | 4 days | Bente et al., 2013[22] |
| Duration of infectiousness in humans | $D_{iH}$ | 9 days | Fillâtre et al., 2019[23] |
| Duration of immunity in humans | $D_{mH}$ | 3650 days | Assumption |
| Fraction of human infection resulting in a clinical case | $\phi$ | 0.31 (CrI 95% 0.28–0.33) | Estimated |
| Proportion of farmers immune at time 0 | $p_F$ | 0.1333 | Mustafa et al., 2011[17] |
| Proportion of others immune at time 0 | $p_O$ | 0.0469 | Mustafa et al., 2011[17] |
| Case fatality rate of CCHF | $CFR_{cchfv}$ | 0.33 | Niazi et al., 2019(16) |
| **Demographics** | | | |
| Livestock population size | $N_L$ | 15,193 | FAO 2008 [24] |
| Livestock ageing factor (1/months) | $\delta$ | 1/12 | Assumption |
| Livestock monthly death rate | $\mu$ | $\mu_a = \begin{cases} 0.0761 & for\, a=1 \\ 0.0743 & for\, a=2 \\ 0.0746 & for\, a=3 \\ 0.0744 & for\, a=4 \\ 0.0747 & for\, a=5 \end{cases}$ | See Fig A in S2 Text |
| Population size—Farmers | $N_F$ | 7,614 | USAID 2008 |
| Population size—Other occupations | $N_O$ | 17,768 | USAID 2008 |
| Life expectancy—humans | $L_H$ | 61.5 years | World bank 2008–2014[25] |
| Monthly birth rate humans | $b_H$ | 1/ (12*61.5) | Assumption |
| Monthly birth rate in livestock | $b_L$ | $\mu$ | Assumption |
| **Viral transmission parameters** | | | |
| Between livestock transmission temperature dependent | A | 0.33 (CrI 95% 0.2–0.4) | Estimated |
| Transmission rate from livestock to farmers | $\beta_F$ | 0.28 (CrI 95% 0.15–0.34) | Estimated |
| Other occupations relative transmission factor(relative to farmers) | O | 0.3 (CrI 95% 0.1–0.5) | Estimated |
| Transmission rate from livestock to other occupations | $\beta_o$ | $O\beta_F$ | Assumption |
| *Vaccination parameters* | | | |
| Vaccine efficacy | $\kappa$ | 90% | Assumption |
| Time to vaccine protection | $D_{pV}$ | 2 weeks | Assumption |

*Estimated values represent the posterior mean and 95% CrI for the best most parsimonious model, i.e., saturation deficit obtained during calibration (see section S3 Text for calibration details).

¥ Livestock age stratification groups where $a$ = 1 reflects 0 to 12 months; $a$ = 2 for 13 to 24 months; $a$ = 3 for 25 to 36 months; $a$ = 4 for 37 to 48 months, $a$ = 5 for 48 months and older

**Table 2. Environmental drivers as surrogate markers of tick activity.**

| Environmental indicator | Description | How we modelled it | Source of data |
|---|---|---|---|
| Soil temperature (ST) (Celsius) | Temperature of the soil in the first layer (0–7 cm) taken at 10:00 AM | Vector of monthly average from April 2008 to January 2019. We assume a tick activity range between 12°C and 30°C. | ERA5 atmospheric variables, centred in a polygon in Herat (ECMWF and Copernicus[26] [27]) |
| Relative humidity (RH) | It is a measure of vapor content in the air. | Vector of monthly average from April 2008 to January 2019. As ticks prefer dry hot weather, we use the complement (1-RH) to indicate increase in tick activity | Constructed from air temperature (T), dew point temperature (Td), and surface pressure from ERA5 (ECMWF and Copernicus[26] [27]) |
| Saturation deficit (SD) | A measure of the drying power of the air. It accounts both for air temperature, vapor pressure and relative humidity. | Vector of monthly average from April 2008 to January 2019. Given that SD includes temperature, we use a simple regression model to find the SD range of tick activity matching the ST range. | ERA5 atmospheric variables centred in a polygon in Herat (ECMWF and Copernicus[26] [27]) |
| Normalised difference vegetation index (NDVI) | Combines satellites signals to estimate the density of green on an area of land. It indicates a combination of rainfall, and land change. | Vector of monthly average from April 2008 to January 2019. | NASA, EarthData (MODIS/VIIRS subsets) for Herat [28] |

 c. 80% farmers

 d. 50% farmers

Each intervention is introduced at year 5 of the simulation, with a linear scale up period of three months. The impact of vaccination is measured as the number of human CCHFV infections averted, and early human deaths averted. To assess efficiency of each approach, we also calculate the ratio of total vaccine courses over human infections averted.

## Results

According to our systematic comparison of environmental and epidemiological drivers, a model with Saturation Deficit as a surrogate indicator of tick activity, and an assumption of increased CCHFV reporting capacity, resulted in the best, most parsimonious model fit when assessed with DIC. **Fig 2** shows model outputs for the best performing model against calibration targets. Interestingly, the DIC estimate was very close (within 5 units) for most environmental drivers (see **Table 3**). Only relative humidity displayed a markedly worst fitting scenario. On the other hand, the baseline assumption of increased reporting capacity of human CCHFV was consistently superior to other epidemiological assumptions like the sustained influx of livestock from high endemic areas (**Table 3**). These results suggest a zoonotic endemic transmission that is well captured by the oscillations in the saturation deficit index. In humans, the spill-over would follow the same trend and more importantly, we estimate that only 31% (CrI 95% 28%-33%) of cases would result in symptomatic disease (see **S5 Fig**).

We compare vaccination campaigns directed to animals only, humans only and also combinations of the two and with different campaign frequency. A summary of the overall impact of different vaccination approaches can be found in **Table 4.** Overall, vaccination strategies targeted to humans display a much larger impact (on human cases and deaths averted) compared to animal vaccination campaigns. Our results also suggest that human vaccination is a more efficient approach, reflected in less vaccine courses per human case averted: a single campaign for 80% livestock requires about 12 fold the number of vaccine courses to prevent one human case compared to a vaccine campaign reaching 50% of humans (**Table 4 and Fig 3**). When we compare campaigns targeted to the overall population vs. farmers it is evident that targeting the high risk groups (farmers) results in higher efficiency (**Fig 3D**). An increase in the frequency of campaigns targeted to livestock displays a larger epidemiological impact, while

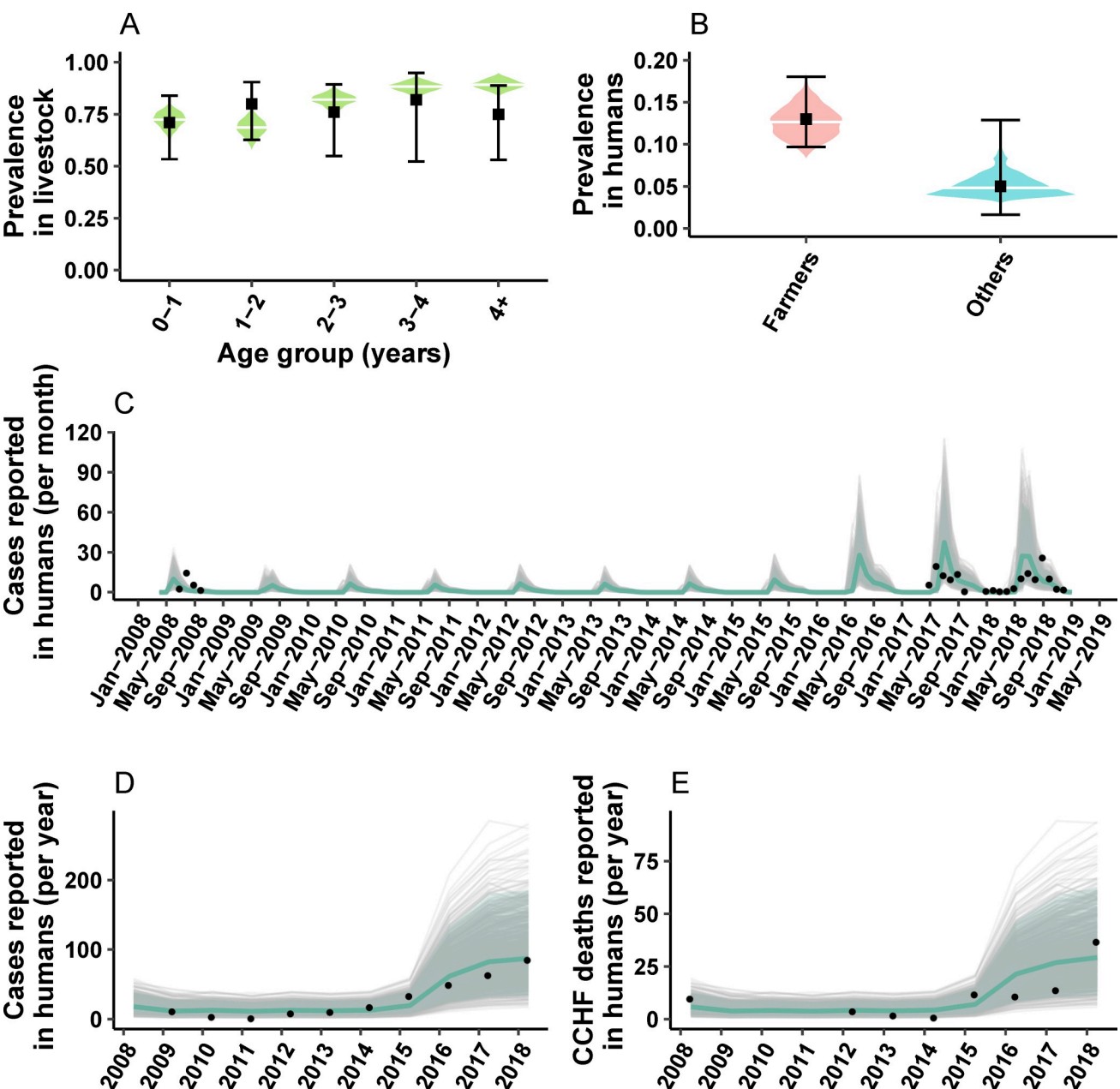

**Fig 2. Model trajectories against calibration target data.** (A) Simulated age stratified CCHFV prevalence among livestock (green density plot), with the median estimate (white horizontal line), against IgG prevalence data for the same age groups as reported by Mustafa et al [17] from Herat (black square shows the mean and error bars the 95%CI). (B) Posterior density and median estimate of IgG prevalence for the population of farmers and other occupations (density plots pink and blue) against IgG prevalence data from Herat reported. We take the prevalence estimate to match the dates of data collection as reported by Mustafa et al. (C) Stochastic model trajectories (grey lines) for monthly incident CCHFV human cases reported in Herat. In shaded pale grey, the 95% CrI and in solid blue, the median estimate. In black dots, monthly incident cases reported in two separate CCHF outbreaks in Herat: in 2008 as reported by Mofleh et al [15], and 2017–2018 as reported by Niazi et al, and Sahak et al [13,16]. (D) & (E) yearly CCHF cases and deaths reported from Herat, against data (black) as reported by Sahak et al, respectively.

resulting in more vaccine courses per case averted over time, compared to a single campaign as seen in **Fig 3B**. Finally, we simulated different combinations of vaccine efficacy and vaccine coverage for vaccination campaigns in humans (**Fig 4**). This analysis shows that for both deaths and infections averted, an intervention targeted to humans yields benefits that are at

**Table 3. Model comparison using DIC.**

| Step 1: Selection of environmental driver assumption | DIC |
|---|---|
| Saturation deficit | 65.21 |
| NDVI | 69.1 |
| Soil temperature | 70.05 |
| Relative humidity | 86.81 |
| Step 2: Selection of Saturation deficit + Epidemiological assumption | |
| Improved reporting | 65.21 |
| Increase influx of livestock (stable reporting) | 77.64 |
| Increase influx of livestock and improved reporting over time | 79.05 |

least one order of magnitude larger compared to the livestock campaign. Importantly, the contour plots show that there is a frontier of high effectiveness that can be reached within a spectrum of combinations of efficacy and coverage.

## Discussion

Understanding the dynamics and epidemiological drivers of transmission are key to establishing priorities for the research and development roadmap for CCHFV. Here we present for the first time a calibrated mathematical model to simulate the transmission of CCHFV in livestock and spill-over into humans in western Afghanistan.

We find that CCHFV in Herat province has reached an endemic state of transmission within livestock, with a yearly cycle that is well reproduced by the oscillations of the saturation deficit index in this geographical area. This index incorporates air temperature and relative humidity, and indicates that the dry and hot months of summer likely result in periods of high tick activity. As long as tick activity data is mostly absent, this approach will continue to be necessary in the future. We highlight the need to extend this analysis to areas where other environmental drivers could be relevant, or where epidemiological factors result in different types of outbreaks. Spill-over transmission into humans mirrors this seasonal pattern, and although stochastic events explain some of the year-to-year variability, the increasing trend in case reporting appears strongly linked to the increased reporting capacity in the country at the time [13]. Importantly, our results show that the volume of spill-over transmission might be much higher than previously expected: we estimate that 31% (CrI 95% 28% - 33%) of transmission events into humans lead to symptomatic disease and therefore to case reporting. Previous

**Table 4. Epidemiological impact of modelled vaccination strategies.** CCHFV Infections and early deaths averted, and number of vaccine courses per clinical case averted, according to the four vaccination scenarios, cumulatively over the period April 2014 to Dec 2018.

| Vaccination scenario | Human CCHF infections averted (CrI 95%) | Human clinical CCHF cases averted (CrI 95%) | Human CCHF deaths averted (CrI 95%) | Total vaccine courses* (CrI 95%) | Vaccine courses per clinical case averted (CrI 95%) |
|---|---|---|---|---|---|
| 80% of livestock in a single campaign | 105 (38–207) | 31 (10–65) | 10 (2–22) | 12,578 (8,857–27,141) | 431 (162–1,438) |
| 80% of livestock yearly | 318 (117–632) | 94 (30–198) | 31 (10–66) | 108,948 (73,236–260,514) | 1,243 (465–4,389) |
| 50% humans in a single campaign | 902 (326–1832) | 266 (88–568) | 87 (28–185) | 9,533 (8,294–10,241) | 35 (16–107) |
| 80% farmers in a single campaign | 686 (270–1039) | 191 (59–490) | 63 (19–164) | 3,700 (3,060–4,242) | 19 (7–62) |

* Cumulative vaccine courses over simulation period

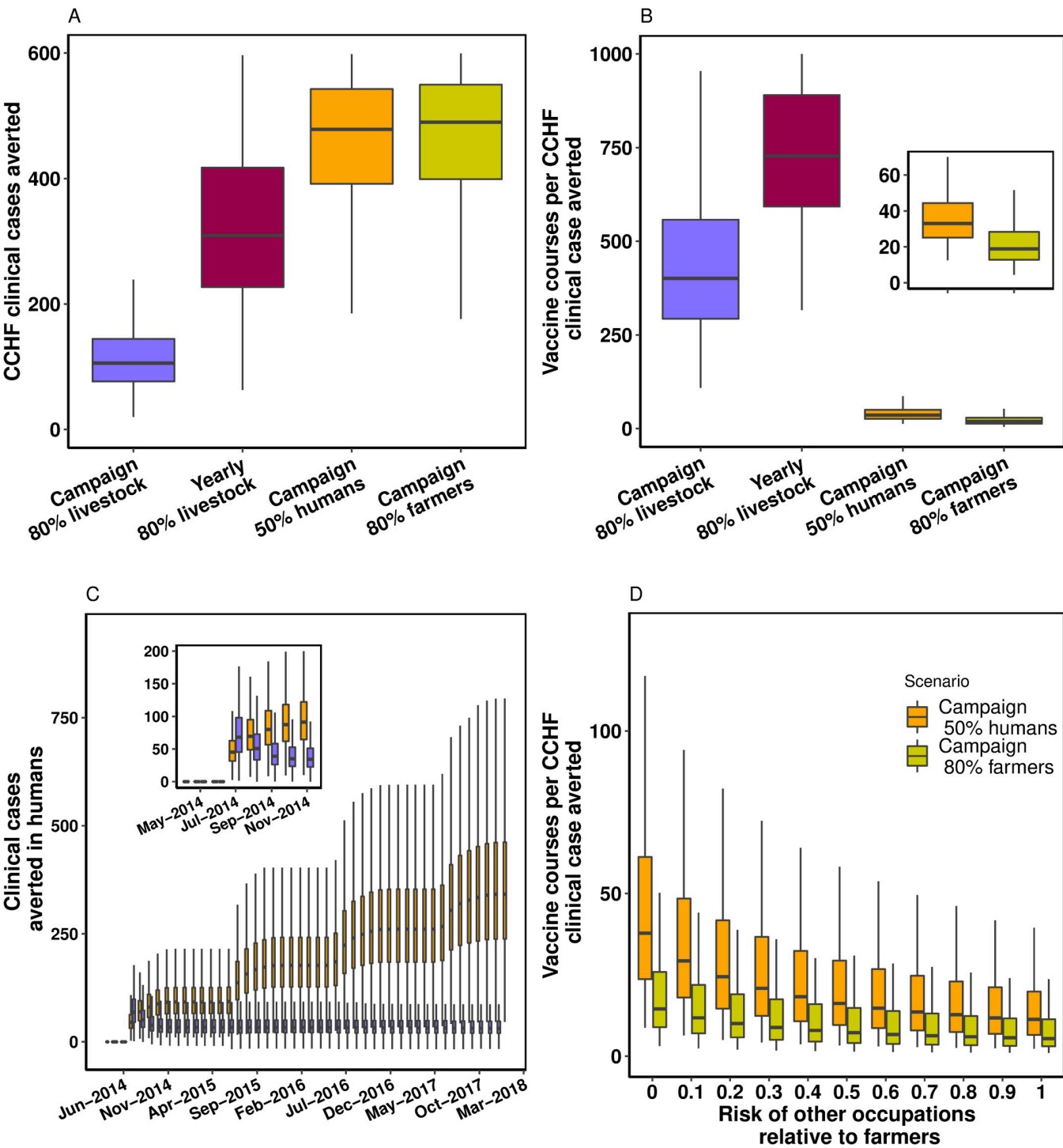

**Fig 3. Population impact of CCHF vaccination strategies.** (A) Cumulative clinical cases averted over 4 years of simulation, for four different scenarios of intervention. (B) the number of vaccine courses required to avert a CCHFV case in humans, estimated as the ratio number of courses over clinical cases averted. The inset window shows a zoom-in for clarity of the two human vaccination interventions. (C), boxplots for the cumulative number of averted clinical cases of CCHFV for the first 4 years of the simulated vaccine period. Inset window shows a zoom-in into the first seven months after vaccination campaigns. In orange, a one-off campaign for vaccinating 50% of humans over a three-month period. In purple, a one-off campaign to vaccinate 80% of livestock over a three-month scale-up period. (D) Sensitivity analysis assessing the effect of the disparity in risk between farmers and other occupations.

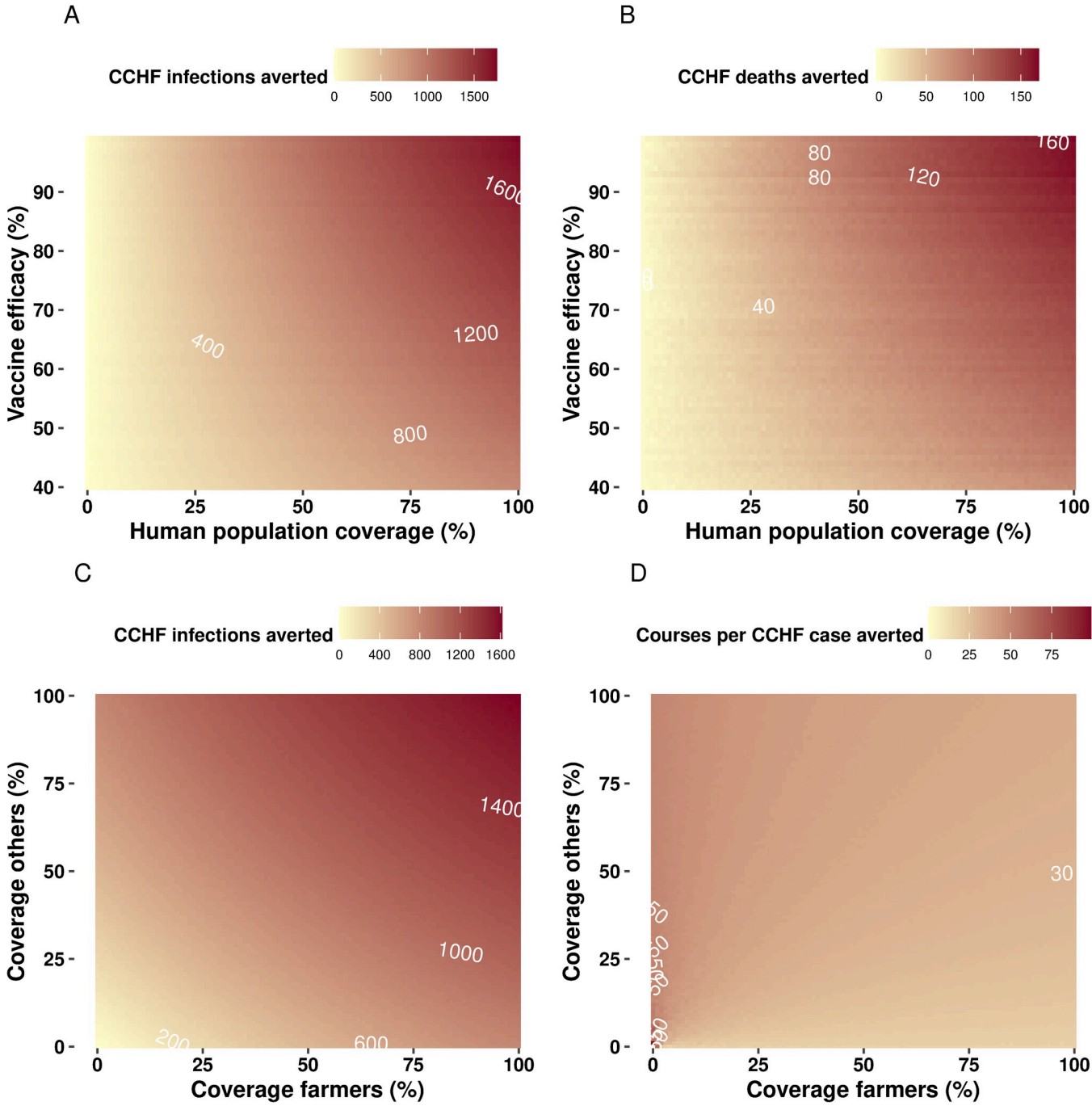

**Fig 4. Exploration of vaccine efficacy and coverage on incidence and mortality reductions.** (A) A contour for combinations of vaccine efficacy and vaccine coverage among humans. Interventions are introduced as a single campaign approach. White solid lines reflect the frontier of effect measured as CCHFV infections averted. (B) shows the combination efficacy vs human vaccination coverage and effect measured as CCHFV deaths averted. (C) and (D), the effect of different levels of coverage among farmers and other occupations on infections averted and doses per case averted, respectively.

evidence from seroprevalence surveys have estimated higher fractions (ranging 88%-100%) of sub-clinical presentation of CCHFV in humans when contrasted to reported cases [29,30]. Our approach to this wide uncertainty in the existing evidence is to try to infer the fraction symptomatic from the calibration process. Nevertheless, further seroprevalence studies in

humans and estimations of the clinical fraction are urgently needed in the field, given the central role this parameter plays in defining the overall burden of disease associated with CCHF and suitable endpoints in future vaccine trials. When estimating the epidemiological impact (in terms of cases and deaths averted) are sensitive to this parameter, as seen in **S6 Fig**.

Future vaccine campaigns against CCHFV might have the largest population impact when applied to humans instead of animals. In the current study setting, immunisation strategies targeted to farmers (the high risk group) are more efficient as they require less vaccine courses per case averted. It is plausible that other epidemic settings with a more concentrated profile of risk could lead not only to more efficient but more impactful targeted interventions. The latter might require further investigation.

Our results exemplify the challenges posed by animal vaccination: livestock campaigns have a rapid impact, but their effect rapidly wanes as livestock population turnover prevents further accumulation of population immunity. More frequent vaccination campaigns increase the long term impact but not enough to match interventions directed to humans. Another challenge that might arise from a livestock vaccination campaign has to do with the fact that asymptomatic CCHFV infection in livestock might result in poor compliance from farmers and animal owners, as immunisation for innocuous infections will not be a priority. Finally, animal vaccination not only shows in our study to return lower benefits, but it also requires a much larger number of vaccine courses per human case averted.

The current study is restricted to one location, and our assessment of environmental drivers could yield different results in different ecological and climatic settings. A natural progression of this analysis would be to assess this modelling approach in other countries where endemic CCHFV transmission is suspected or established. Another limitation directly related to this is the lack of data on tick activity and tick abundance for this setting. Such data are rare not only in this context but in any setting around the world. Furthermore, by not incorporating an explicit tick-vertebrate or tick-human mechanism in the model there is a possibility that other factors affecting tick populations or tick activity could result in different epidemic trajectories. However, a strength of our analysis is the solution we provide by systematically testing environmental surrogates for tick activity using climatic factors that are well known predictors of tick activity [31]. There is a clear need to collect and analyse tick activity in addition to wildlife host data to better understand the drivers of CCHFV transmission [32].

Our approach focuses on the transmission into humans from human-animal contact, and we ignore human-to-human transmission. Previous evidence shows that human-to-human transmission is plausible and nosocomial transmission has been reported before [33,34]. However, we expect this aspect of transmission to contribute marginally to the annual reported trends of CCHFV cases in Afghanistan as most infections arise outside the hospital environment and are linked to animal handling activities [15,16].

Our assumption on transmission is also central for interpreting the impact of vaccination campaigns, since the absolute epidemiological impact is necessarily limited by the population size of the targeted human group. For this reason, a measure of efficiency like doses per case averted might be a better indicator of intervention performance.

In this work we also ignore transmission cycles in wildlife. This can be important for maintaining more stable levels of endemicity, but the absence of data prevents us from designing a more complex transmission network for Herat.

In conclusion, CCHFV is likely to be endemic in western Afghanistan, with a seasonal pattern which is robustly predicted by climatic factors which we explore in this work. The increasing number of human cases reported in Herat are most likely explained by increasing trends in reporting capacity in the country, and more importantly, these cases are reflecting only a fraction of the overall volume of human infection. Vaccination campaigns in humans are

more impactful and efficient in the medium and long term compared to livestock vaccination. Finally, targeting campaigns to groups with an increased risk of infection, like farmers, are the most efficient strategies in our assessment and should be a key component of future vaccine implementation roadmaps for CCHFV.

## Supporting information

**S1 Fig. Model structure for the transmission of CCHFV amongst livestock.** The structure shows the compartments and state transitions for livestock. Births occur at a rate equal to the mortality rate to maintain a population at equilibrium. Mortality rates are estimated to achieve a known livestock age-distributed population (see S2 Text). Offspring from prevalent CCHFV animals acquire transient immunity at birth through first colostrum ($R_i$). This immunity lasts for an assumed average period of 6 months. After this period, livestock move to the susceptible stage ($S_1$). Susceptible livestock ($S_a$) acquire CCHFV with a force of infection $\lambda_L$ that leads to an infectious period ($I_a$) with mean duration $D_{iL}$, expressed as inverse time rate $D_{iL}^{-1}$. We assume that livestock lose immunity at a rate $D_{mL}$. Vaccination is implemented by recruiting susceptible animals at a rate $v(t)$. The effective number of immunised livestock is also defined by the efficacy of the vaccine ($\kappa$). At first, vaccination does not confer immunity $V_a$, which is only acquired after a period of length $D_{pV}$. In this structure, subscript $a$ points to the age category within the age structure, over which transitions occurs as depicted with the shaded grey structure in the background
(TIF)

**S2 Fig. Model structure for the spillover transmission of CCHFV and disease progression in humans.** Humans are born at a rate reflecting the life expectancy in Afghanistan (keeping population size constant in the absence of infections), and split into the two human categories considered in this model. Namely farmers (the high-risk group) and other occupations. This distribution is taken from previous USAID surveys in the country (see parameters Table 1 in the main text). The categorisation by occupation in the model is reflected in this structure and in the mathematical equations using subscript $k$ (0 = farmer; 1 = others). Infection is acquired in humans with force of infection $\lambda_k$, with differential risk $k$. Infection is followed by a latent period $\hat{E}_k$ with mean duration $D_{lH}$ that leads to an infectious period $\hat{I}_k$. This infectious period can lead to either recovery $\hat{R}_k$ or death. Death from CCHFV in humans is described in the model as the competing hazard $\mu_{iH}$ that summarise the case fatality ratio for CCHFV. We assume waning immunity in humans that leads back to susceptible stage at a rate $D_{mH}^{-1}$. Vaccination occurs at rate $v(t)$, differential by occupation. Effective number of immunised people is finally defined by the efficacy of the vaccine ($\kappa$). Vaccine protection occurs after vaccination after a mean period $D_{pV}$.
(TIF)

**S3 Fig. Model trajectories against calibration target data for a model with lifelong immunity protection among livestock.** Panel A shows the age stratified simulated CCHFV IgG prevalence among livestock (green density plot), with the median estimate (white horizontal line), against IgG prevalence data for the same age groups as reported by Mustafa et al [14] from Herat (black square shows the mean and error bars the 95%CI). Panel B shows the posterior density and median estimate of IgG prevalence for the population of farmers and other occupations (density plots pink and blue) against IgG prevalence data from Herat reported. We take the prevalence estimate to match the dates of data collection as reported by Mustafa et al. Panel C shows stochastic model trajectories (grey lines) for monthly incident CCHFV human cases reported in Herat. In shaded pale grey, the 95% CrI and in solid blue, the median

estimate. In black dots, monthly incident cases reported in two separate CCHF outbreaks in Herat: in 2008 as reported by Mofleh et al [16], and 2017–2018 as reported by Niazi et al, and Sahak et al [15,17]. In Panels D and E, yearly CCHF cases and deaths reported from Herat, against data (black) as reported by Sahak et al.
(TIF)

**S4 Fig. Deviance information Criterion DIC values for different modelling assumptions on environmental driver, case reporting assumption, and duration of livestock acquired immunity.** As mentioned in the main text, saturation deficit with baseline assumptions about reporting produces the lowest DIC (best model fit). Importantly, a model with lifelong immunity among livestock shows the worst performance.
(TIF)

**S5 Fig.** Transmission dynamics of CCHF in Herat, Afghanistan 2008–2018 In panel A, simulated trajectories of monthly CCHF incidence in a spectrum from reported clinical cases (blue shade), to all clinical cases (green) and all cases (red) including symptomatic/subclinical cases. The shaded area shows the 95% CrI. In Panel B, the simulated effective reproduction number for CCHFV in livestock. These are results for the final selected model, i.e., "saturation deficit driver" model.
(TIF)

**S6 Fig. Effect of clinical fraction on the epidemiological impact of human vaccination.** Increase in vaccination coverage among humans results as expected in increased epidemiological impact, in infections (panel A), clinical cases (panel B) and deaths (panel C). We take extreme values for the clinical fraction parameter to show that a scenario where only 5% of infections are symptomatic (blue boxplots) results in a drastic decline in epidemiological impact in cases and deaths when compared against a highly symptomatic scenario (green boxplots). The baseline calibrated model is shown in orange, for which the mean clinical fraction is 31%.
(TIF)

**S1 Text. Model equations; Table A in S1 Text. Age distributed CCHFV prevalence among livestock; Fig A in S1 Text. Polynomial model prediction model of Saturation deficit on Air temperature.**
(DOCX)

**S2 Text. Livestock demographic model; Fig A in S2 Text. Age distribution in cattle into 5 age yearly groups.**
(DOCX)

**S3 Text. Model calibration; Table A in S3 Text. Calibration target datasets for CCHFV in Herat, Afghanistan; Fig A in S3 Text. MCMC Trace plots; Fig B in S3 Text. Density plots; Fig C in S3 Text. Gelman-Rubin diagnostic.**
(DOCX)

**S4 Text. Environmental drivers.**
(DOCX)

## Author Contributions

**Conceptualization:** Juan F. Vesga, W. John Edmunds, Raphaëlle Métras.

**Data curation:** Edris Ayazi, Toby Leslie.

**Formal analysis:** Juan F. Vesga, Madeleine H. A. Clark, Andrea Apolloni, Raphaëlle Métras.

**Funding acquisition:** W. John Edmunds.

**Investigation:** Juan F. Vesga, Madeleine H. A. Clark, Andrea Apolloni, Toby Leslie.

**Methodology:** Juan F. Vesga, W. John Edmunds, Raphaëlle Métras.

**Writing – original draft:** Juan F. Vesga, W. John Edmunds, Raphaëlle Métras.

**Writing – review & editing:** Juan F. Vesga, Madeleine H. A. Clark, Edris Ayazi, Andrea Apolloni, Toby Leslie, W. John Edmunds, Raphaëlle Métras.

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
