## [Decision Letter · Decision Letter 0]

21 Mar 2022

Dear Dr Vesga,

Thank you very much for submitting your manuscript "Transmission dynamics and vaccination strategies for Crimean-Congo haemorrhagic fever virus in Afghanistan: a modelling study" for consideration at PLOS Neglected Tropical Diseases. As with all papers reviewed by the journal, your manuscript was reviewed by members of the editorial board and by several independent reviewers. The reviewers appreciated the attention to an important topic. Based on the reviews, we are likely to accept this manuscript for publication, providing that you modify the manuscript according to the review recommendations. 

The reviewers all agreed that this manuscript is valuable and only needs a few minor clarifications/corrections before publication. Reviewer 1 commented on the source of some assumptions made in the model, so please be sure the assumptions are well-explained and well-cited. The remaining comments from reviewers are primarily editorial changes. Please consider them in your revision.

Sincerely,

Brianna R Beechler, Ph.D., DVM

Associate Editor

Jeremy V. Camp, PhD

Deputy Editor

The reviewers all agreed that this manuscript is valuable and only needs a few minor clarifications/corrections before publication. Reviewer 1 commented on the source of some assumptions made in the model, so please be sure the assumptions are well -explained and well-cited. The remaining comments from reviewers are primarily editorial changes. Please consider them in your revision.

Reviewer's Responses to Questions

**Key Review Criteria Required for Acceptance?**

**Methods**

-Are the objectives of the study clearly articulated with a clear testable hypothesis stated?

-Is the study design appropriate to address the stated objectives?

-Is the population clearly described and appropriate for the hypothesis being tested?

-Is the sample size sufficient to ensure adequate power to address the hypothesis being tested?

-Were correct statistical analysis used to support conclusions?

-Are there concerns about ethical or regulatory requirements being met?

Reviewer #1: The objectives of the study were clearly stated in the author summary and introduction of the manuscript. The authors state that the study is to develop an analytical model looking at different ecological and epidemiological variables related to CCHF to shed light on its transmission dynamics in humans and what effect vaccination of different populations may have on these dynamics. 

The study design was a mathematical model analyzing various inputs related to ecological and epidemiological factors for CCHF virus and was appropriate given the stated objectives. The methodology and inputs were appropriate and reasonable given the study questions being addressed. There was a comprehensive review of various inputs for all aspects and stages of the livestock, human and vector components of CCHF transmission cycles. There were a number of inputs and values that were “estimated” and it was not always clear how these inputs and values were determined.

A few clarifications related to the analysis and variables and values used in the model analyses: 

• It was not clear if the authors accounted for varying contact rates with livestock between farmers and the general population. The calibrated model shows the proportion of contact for other occupational groups was 0.3 and transmission from livestock to farmers was 0.28. Was this assumption based on farmers having higher pre-existing CCHF immunity? 

• The authors estimate waning CCHF immunity on livestock after 5 years. What was this based on? And even if waning immunity, by what factor would this reduce protection against re-infection. 

• The model assumes these animals after 5 years are able to be re-infected, but the general consensus for CCHF is infection confers lifelong immunity for both animals and humans. 

• The human immunity is also estimated at 10 years, and we presume lifelong immunity, or at least protective immunity even if waning measurable antibody titers. 

• The authors estimate the fraction of human CCHF infections that result in clinical cases was .31. Based on first-hand experience and study this seems very low and could be an underestimate. 

• These assumptions may impact the model outputs if not validated or evidence is available for documented re-infection of humans and/or animals 

• Authors state ticks bury in soil once temp goes beyond 30 degrees. What is the shape of the function for declining transmission? Is it linear? Exponential? What assumptions are made? Was this determined by model calibration? 

No ethical concerns were noted.

Reviewer #2: Yes the objectives are clear on the design appropriate to achieve the objectives. Also correct statistics has been done but this would be checked by a statistician or someone who is a stronger mathematical background to make sure that today models are aware formulated and calibrated

Reviewer #3: None.

**Results**

-Does the analysis presented match the analysis plan?

-Are the results clearly and completely presented?

-Are the figures (Tables, Images) of sufficient quality for clarity?

Reviewer #1: The analysis plan described in the manuscript of the mathematical model was appropriate and straight forward. All analysis and results were clearly described and the relevant findings were highlighted. The results appear to be reasonable and as expected based on the assumptions, inputs and general epidemiology and impact of vaccination on the proposed target populations. 

The tables and figures accurately represent the data presented and are thorough and detailed.

Reviewer #2: The results are represented and well supported by the supplementary material that gives the details of some of the analysis and their their results

Reviewer #3: The results are clearly and completely presented. The figures and tables are of sufficient quality.

**Conclusions**

-Are the conclusions supported by the data presented?

-Are the limitations of analysis clearly described?

-Do the authors discuss how these data can be helpful to advance our understanding of the topic under study?

-Is public health relevance addressed?

Reviewer #1: The overall discussion and conclusions stated by the authors agrees with the data and analysis presented. In general the authors discuss their model findings and the impact of vaccination on the estimated transmission of CCHF. A few items requiring further clarification:

• The first point of discussion highlights the lack of tick vector data in which to calibrate and use for the model. It is not stated in the manuscript what the assumption of percent of total ticks are infected with CCHF and how this impacts the potential infectivity of cattle/human populations. From practical experience tick infectivity could be low, but the documented seropositivity in livestock is often very high. Therefore the estimate for livestock-livestock CCHF transmission could be happening at a higher rate than estimated in the model, although the estimates used could offset each other and therefore not have any impact on this model or effect any of the resulting outputs. 

• The authors state again that an estimated 31% of spillover cases become symptomatic and therefore detectible by surveillance. As stated above, this seems too low of an estimate. The two references (29, 30) showing that the potential for asymptomatic infection for CCHF could be higher. I believe the authors may have misinterpreted these references and greatly overestimated the number of potential asymptomatic infections. One of the references use a serological assay known for its overestimation of the true IgG seropositivity. The second reference is a validation study of IgG positive CCHF cases from three independent serosurveys where the IgG seropositivity that could be attributable to asymptomatic CCHF as less than 5%. 

Generally, the effects of vaccination on the targeted populations and the resulting reduction in estimated CCHF human cases is as would be expected given the analysis. The only change to the results could be the absolute number calculated from the model if some of the assumptions were to be adjusted up or down based on more real-world estimates.

Reviewer #2: The study is a good public health importance because it touches and emerging disease that has not been given attention by the public health bodies. A disease that causes mortality and mobility and no vaccine is in place yet to be used especially in high-risk groups such as farmers and Herdsmen. However more work would be needed to implement some of these recommendations in the article. One research or one publication is not enough until some of the work is done in the different countries where CCHF is prevalent so as to give a comprehensive picture. The authors could edit their recommendations and conclusions accordingly.

Reviewer #3: The conclusions are well supported by model results. The limitations, particularly around the large gap in information for CCHF, are well described. The results are contextualized well and the public-health relevance is well addressed.

**Editorial and Data Presentation Modifications?**

Reviewer #1: none

Reviewer #2: The manuscript could receive another read and make sure that the English drama is reading well especially in the abstract. For example I do not know the use of the word"course "as used in this abstract or are they meaning "Cases".

Reviewer #3: Main text

------------

P 7: The formula at the end of Figure 1 caption should have N_L instead of N, like the equation on line 136.

P 7: Lines 120—125 should continue the caption of Figure 1.

P 10: What is the value of the monthly birth rate in livestock?

P 10: The notation for transmission from livestock to other occupational groups should be \\beta_O.

P 10: What is the value of the time to vaccine protection?

P 10: Lines 176–178 are an odd place for the ethics statement. Perhaps move it elsewhere.

P 17: On line 251, "panel B of Figure 3B" should be "Figure 3B" in parallel with "Figure 3D" on line 258.

Supplement

----------

P 4–6: What are the initial conditions for the livestock model? How long is period of time is simulated?

P 4: The second line of Eq. 1 is not rendered correctly.

P 4: Comparing Eq. 2 to Figure S1, the term for births from recovered livestock (R_a(t)) is missing.

P 5: In Eq. 3, the "L" following lambda should be a subscript.

P 5: In Eq 7, for Z_{i,j}, the lower left term Z_{5,5} should be 0, not -1, as the animals 5th age group do not age out, only leave through mortality (mu_a).

P 6–8: What is the time step used in the human model? What are the initial conditions for the human models? How long is period of time is simulated?

P 8 & 9: I don't understand how saturation deficit, air temperature, and tick activity are related from the section and Figure S3. What are the black dots in the figure? Please explain further.

P 10: I found "with the last categories being those 4+ years" to be confusing. I recommend "except the last age group includes all those aged 4+ years" or similar.

P 10: "It has been observed that Hyalomma spp. similarly to other tick species, are strongly driven in their reproduction cycles and also feeding activity by different climatic and land composition variables." Please provide citations at the end of this sentence.

P 11: "prevalent I" should be "prevalence".

P 12: "As a rule of thumb, values below 1.1 are typically considered to indicate convergence." Please provide a citation, perhaps to Gelman's book.

P 12: You refer to the test statistic (R with two dots) as the "scale reduction factor" then the "Potential scale reduction factor (PSRF)" in the same paragraph, with the latter also used in Figure S7. Please either use "potential" throughout or explain why "potential" is added later.

P 12: "shoes" should be "shows".

**Summary and General Comments**

Reviewer #1: Overall, this manuscript is a good first analysis of the impact vaccination could have on human CCHF infections given the transmission dynamics in a specific geographic location in Afghanistan. It may not be as generalizable given the dynamics of CCHF varies so widely across different geographic locations and ecological niches. Some limitations the authors properly address, but lack of location specific variables/data points could bias their analysis if taken from locations distant from the study location (e.g CCHF data from Turkey) to help validate or calibrate the model.

Reviewer #2: This manuscript if published would you give a body of scientific knowledge on how to model infectious diseases especially vector- born infectious diseases. So it's of good public health importance and I would recommend it gets published. It also presents a challenge which is mentioned as a limitation for not getting tick activity data and using environmental data as a proxy which is a challenge that we see in most of the modeling studies involving vector dynamics. The authors could also mention the practicability of the recommendations in the study country. How easy is it to vaccinate cattle.animals, farmers or high risk in study country. If these recommendations taken on; are they practical.

Reviewer #3: The manuscript presents a novel modeling study of Crimea–Congo hemorrhagic fever (CCHF) in Afghanistan that provides a solid footing for important and relevant findings. The model used has an appropriate balance of simplicity and use of available data, which is particularly important due to the complex transmission cycle of CCHF and complex life cycle of its tick vectors. Key missing information was estimated using Bayesian methods and model selection, which is appropriate.

Model selection showed that saturation deficit was the most important environmental driver of infections (through its effect on ticks) and that recent increases in incidence were best explained by improved reporting, rather than an increase in livestock imports. The latter seems like an important finding, although it was not heavily emphasized.

Most importantly, the work shows that, for a hypothetical vaccine, vaccinating humans is much more effective at reducing infections, clinical cases, and deaths in humans than vaccinating livestock, and vaccinating farmers, who at high risk due to contact with livestock, is particularly efficient per dose of vaccine. These are important and relevant findings for mitigating the impact of this disease.

PLOS authors have the option to publish the peer review history of their article (what does this mean?). If published, this will include your full peer review and any attached files.

Reviewer #1: No

Reviewer #2: Yes: Luke Nyakarahuka

Reviewer #3: Yes: Jan Medlock

Figure Files:

Data Requirements:

Reproducibility:

References

---

## [Decision Letter · Decision Letter 1]

29 Apr 2022

Dear Dr Vesga,

We are pleased to inform you that your manuscript 'Transmission dynamics and vaccination strategies for Crimean-Congo haemorrhagic fever virus in Afghanistan: a modelling study' has been provisionally accepted for publication in PLOS Neglected Tropical Diseases.

Best regards,

Brianna R Beechler, Ph.D., DVM

Associate Editor

Jeremy V. Camp, PhD

Deputy Editor

Editorial Note: The reviewers are happy with your revisions. Reviewer 3 noted that a few equations did not render correctly in the most recently submitted version so be sure to check the proofs carefully to ensure equations are showing correctly.

Reviewer's Responses to Questions

**Key Review Criteria Required for Acceptance?**

**Methods**

-Are the objectives of the study clearly articulated with a clear testable hypothesis stated?

-Is the study design appropriate to address the stated objectives?

-Is the population clearly described and appropriate for the hypothesis being tested?

-Is the sample size sufficient to ensure adequate power to address the hypothesis being tested?

-Were correct statistical analysis used to support conclusions?

-Are there concerns about ethical or regulatory requirements being met?

Reviewer #1: Methods have been clarified. Revisions are sufficient for acceptance

Reviewer #3: (No Response)

**Results**

-Does the analysis presented match the analysis plan?

-Are the results clearly and completely presented?

-Are the figures (Tables, Images) of sufficient quality for clarity?

Reviewer #1: Questions related to results have been addressed by reviewers. Although not completely able to be addressed, the limitations of this have also been highlighted and readers are aware.

Reviewer #3: (No Response)

**Conclusions**

-Are the conclusions supported by the data presented?

-Are the limitations of analysis clearly described?

-Do the authors discuss how these data can be helpful to advance our understanding of the topic under study?

-Is public health relevance addressed?

Reviewer #1: Conclusions are sufficient for accceptance

Reviewer #3: (No Response)

**Editorial and Data Presentation Modifications?**

Reviewer #1: (No Response)

Reviewer #3: I appreciate the authors' attention to the minor issues I raised in the previous round of reviews. Except as noted below, the authors resolved the issues I raised.

Many of the equations in the supplement, which was only available as DOCX, did not render at all for me. I believe this was some software compatibility problem. Thus, I was unable to confirm the issues from my first review that are listed below were resolved.

Supplement

----------

P 4: The second line of Eq. 1 is not rendered correctly.

P 4: Comparing Eq. 2 to Figure S1, the term for births from recovered livestock (R_a(t)) is missing.

P 5: In Eq. 3, the "L" following lambda should be a subscript.

P 5: In Eq 7, for Z_{i,j}, the lower left term Z_{5,5} should be 0, not -1, as the animals 5th age group do not age out, only leave through mortality (mu_a).

**Summary and General Comments**

Reviewer #1: None. Nice first paper to assess the impact of vaccination on CCHF

Reviewer #3: (No Response)

PLOS authors have the option to publish the peer review history of their article (what does this mean?). If published, this will include your full peer review and any attached files.

Reviewer #1: No

Reviewer #3: **Yes: **Jan Medlock

---

## [Editor Report · Acceptance letter]

18 May 2022

Dear Dr Vesga,

We are delighted to inform you that your manuscript, "Transmission dynamics and vaccination strategies for Crimean-Congo haemorrhagic fever virus in Afghanistan: a modelling study," has been formally accepted for publication in PLOS Neglected Tropical Diseases.

Best regards,

Shaden Kamhawi

co-Editor-in-Chief

Paul Brindley

co-Editor-in-Chief
